# Genetic Variation in Water-Use Efficiency (WUE) and Growth in Mature Longleaf Pine

Ana C. Castillo [1], Barry Goldfarb [1,*], Kurt H. Johnsen [2], James H. Roberds [3] and C. Dana Nelson [4]

[1]  Department of Forestry and Environmental Resources, North Carolina State University, Raleigh, NC 27695-8008, USA; accasti2@ncsu.edu

[2]  USDA Forest Service, Southern Research Station, 1577 Brevard Road, Asheville, NC 28806, USA; kjohnsen@fs.fed.us

[3]  USDA Forest Service, Southern Research Station, 23332 Success Road, Saucier, MS 39574, USA; jroberds@fs.fed.us

[4]  USDA Forest Service, Southern Research Station, 730 Rose Street, Lexington, KY 40546, USA; dananelson@fs.fed.us

*  Correspondence: bgg@ncsu.edu, Tel.: +1-919-515-4471

**Abstract:** The genetic and physiological quality of seedlings is a critical component for longleaf pine (*Pinus palustris* Mill.) restoration, because planting genetic material that is adapted to environmental stress is required for long-term restoration success. Planting trees that exhibit high water-use efficiency (WUE) is a practice that could maximize this species' survival and growth in a changing climate. Our study evaluates genetic variation in WUE and growth, as well as WUE-growth relationships, a key step to determine potential for breeding and planting trees with high WUE. We measured carbon isotope discrimination ($\Delta$)—a proxy for WUE—in 106 longleaf pine increment cores extracted from trees belonging to nine full-sib families. Tree diameter and total tree height were also measured at ages 7, 17, 30 and 40 years. Each increment core was divided into segments corresponding to ages 7–17, 18–30 and 31–40, representing early, intermediate and mature growth of the trees. We identified significant genetic variation in DBH and WUE among families that merit further exploration for identifying trees that can potentially withstand drought stress. Mean family growth rates were not associated with mean family values for carbon isotope discrimination. Family variation in both diameter growth and WUE but no relationship between family values for these traits, suggests it is possible to improve longleaf pines in both diameter growth and WUE through appropriate breeding.

**Keywords:** *Pinus palustris*; carbon isotope discrimination; family variation; climate change; reforestation

## 1. Introduction

Longleaf pine (*Pinus palustris* Mill.) once dominated the coastal plains of the southeastern United States, covering 25 million hectares but its area has been reduced to less than 3% of its range prior to European colonization of North America [1]. Extensive efforts to restore longleaf pine ecosystems have been initiated across its native range, which extends from southeastern Virginia to eastern Texas, as approximately 53,000 hectares of longleaf pine were established in 2017 alone [2]. Despite high levels of longleaf pine plantation establishment, there is limited knowledge regarding the genetic make-up and subsequent physiological characteristics of the seedlings of this species being planted across the southeastern US.

Compared to the other, more commercially valuable, species such as loblolly pine and slash pine (*Pinus taeda* L. and *Pinus elliottii* Engelm., respectively), there has been less effort toward developing

and deploying genetically improved planting stocks of longleaf pine. Previous studies, however, have determined that the amount of genetic variation in longleaf pine is comparable to that found in other southern pines, making it a suitable candidate for genetic improvement [3,4].

Furthermore, longleaf pine trees may live to an age greater than 400 years and are likely to experience substantial climate variability, especially in the face of climate change. Climate models predict increased temperatures for the southeastern US and decreased water availability for the Gulf coast and western regions of the longleaf pine range (Louisiana, Mississippi, Alabama, western Georgia and the Florida panhandle) [5]. One potential strategy to adapt forests to future climates is to plant genotypes that use less water for biomass production and can better cope with a decrease in water availability [6]. Longleaf pine trees with good growth and moderate water loss, that is, high water-use efficiency (WUE), would potentially be better adapted to future climates.

At the leaf level, intrinsic WUE is the ratio of the rates of photosynthesis and stomatal conductance [7]. WUE is negatively correlated with carbon isotope discrimination ($\Delta$), the ratio of carbon 13 and carbon 12 isotopes ($^{13}C/^{12}C$) resulting from isotope fractionation of $^{13}C$ by plants [8]. Photosynthetic enzymes discriminate against the $^{13}C$ isotope during photosynthesis, so that $^{13}C$ incorporated into plant tissue is less than that found in the atmosphere, especially when stomatal conductance is low. Carbon isotope discrimination by plants is regarded as a reliable proxy for intrinsic WUE of photosynthesis as evidenced by numerous empirical studies reviewed in Moran et al. [9]. The $\Delta$ signature is imprinted on photo-assimilates, which are translocated from the source leaves to cells in cambial tissue and to wood constituents, thus storing the $\Delta$ signature in tree-rings [10]. As a result, unless there is a substantial amount of stored carbon used later for wood production, differences observed in the $\Delta$ signature across tree rings represent the outcome of physiological processes influenced by growing conditions at the time cellulose, lignin and hemicellulose constituents are formed [11].

Genetic variability in $\Delta$ and its relationship to growth traits has been studied in a variety of tree species to investigate the feasibility of breeding trees for high WUE. Narrow-sense heritability estimates for $\Delta$ in tree populations have been found to vary greatly, ranging from $h^2 = 0.09$ in loblolly pine seedlings [12] to $h^2 = 0.72$ in hoop pine (*Araucaria cunninghamii* Aiton ex A.Cunn.) [13]. The relationship between carbon isotope discrimination and tree growth has also been studied in some species but results have been variable. Marguerit et al. [6] found moderate narrow-sense heritability for $\Delta^{13}C$ ($h^2 = 0.29$) in maritime pine (*Pinus pinaster* Ait.) and their results indicate that high WUE is not genetically correlated with growth. They concluded it should be possible to select genotypes for both high WUE and superior growth rate. In contrast, Xu et al. [14] found strong, negative genetic correlations ($r = -0.83$ to $-0.96$) between $\Delta^{13}C$ and tree growth, particularly tree height, in clones of an F1 hybrid of slash pine (*Pinus elliottii* Engelm.) and Caribbean pine (*Pinus caribaea* Morelet). Furthermore, in tests across sites that varied in water availability, Johnsen et al. [15] found $\Delta^{13}C$ to be moderately to highly heritable (family mean $h^2 = 0.54$) and strongly, negatively correlated with growth ($r = -0.97$) in a black spruce (*Picea mariana* Mill.) population. Moreover, in an earlier study consisting of a subset of four families included in the same black spruce tests, a negative correlation between $\Delta$ and tree height was found only in data collected from the driest sites [16]. Similarly, in an investigation of WUE in five-year-old longleaf pine, a negative relationship between tree height and $\Delta^{13}C$ was observed ($r = -0.55$) only on an experimental site that had not been irrigated and was found to have the lowest productivity [17]. Additional research is needed to determine the relationship between water use efficiency and growth in longleaf pine, especially in older trees and across life stages.

In order to ensure the adaptability of seed used for restoration of longleaf pine ecosystems, fundamental knowledge of genetic variability existing in populations of this species is needed. In this study, we evaluated full-sib family variation for WUE and growth traits, as well as WUE-growth relationships across different life stages in mature trees from a population of longleaf pine. We hypothesized that tree mean diameter at breast height (DBH) and mean total tree height would differ significantly among families. We also predicted we would observe significant family variation for WUE,

as estimated by Δ measured from wood cores. Finally, we hypothesized that we would find a negative relationship between growth rates and Δ during early, intermediate and mature growth stages.

## 2. Materials and Methods

Measurements were recorded for growth traits and Δ values were obtained from trees growing in plots of a longleaf pine genetic field experiment established in 1960 on the U.S. Forest Service's Harrison Experimental Forest (HEF), located near Saucier, Mississippi, USA (30.65 N, 89.04 W, elevation 50 m) [18]. The HEF spans 1662 hectares within the De Soto National Forest and is representative of about 12.5 million hectares of land with similar soil properties and topography of the southeastern U.S. [19]. The climate in this region is temperate-humid subtropical, with annual temperatures ranging from −7 to 37 °C and a mean annual precipitation of 1651 mm and rainfall distributed relatively evenly throughout the year [19]. Soils on the HEF are well-drained, fine-sandy loams of the Ruston and McLaurin series and are generally low in mineral nutrients.

Trees selected for measurement are individuals in full-sib family plots that make up a subset of the collection of plots included in a 13-parent, partial diallel, field experiment established in 1960 for the purpose of studying trait inheritance patterns and parent-progeny correlations in longleaf pine [17]. The test was planted with seedlings from crosses among parents in the local area of the test. The test was planted in a hexagonal design with each tree 3.66 m from its six nearest neighbors. Mechanical weed control was performed for two years following planting and the test was mowed occasionally thereafter. The Bordeaux mixture fungicide was applied to the seedlings three times a year for the first three years, to minimize the effects of brown spot needle disease. The resulting subset of plots yields an experimental design structure consistent with a balanced, randomized, complete-block design with four blocks and a family structure formed by crossing six parents according to a pattern produced by a partial diallel mating design that results in nine full-sib families (Table 1). Because of tree mortality that occurred after the experiment was established and for purposes of this investigation, we chose to limit our attention to families for which there were at least three living trees per family plot in each experimental block. Nevertheless, only 107 trees were available for data collection, as a result of the mortality of one tree in family 30 × 39. Tree DBH and total tree height were measured at ages 7, 17, 30 and 40 years following establishment of the experiment, with the exception of tree height for at age 30.

**Table 1.** Mating design for the nine longleaf pine families included in our study.

| Male Parents | Female Parents | | | | | |
|:---:|:---:|:---:|:---:|:---:|:---:|:---:|
| | 21 | 26 | 29 | 30 | 32 | 39 |
| 21 | | | | | | |
| 26 | × | | | | | |
| 29 | × | | | | | |
| 30 | | × | × | | | |
| 32 | | × | × | | | |
| 39 | × | | | × | × | |

The × denotes crosses made.

Increment cores were withdrawn from trees at approximately 1.2 m above ground level using a 40.64 cm long, 5.15 mm diameter increment borer bit. Cores were collected from the west side of tree boles during October 2014 and November 2015. Following extraction, the core samples were placed in plastic beverage straws and immediately placed on ice then subsequently stored in a freezer at −30 °C until being shipped on ice to the USDA Forest Service Southern Research Station (SRS) lab at Research Triangle Park, North Carolina, USA.

Segments corresponding to ages 7–17, 18–30 and 31–40 were cut from both halves of each core with a razor blade. The two segments representing each age class were then combined to form a sample for each time period for each tree. Secondary, more mobile, compounds such as resin, lipids, organic acids,

terpenes, phenols and waxes were removed using a solvent solution to avoid variation introduced by their slightly different isotopic composition [20]. First, each sample was dried overnight in a 105 °C oven. After drying, a Soxhlet extractor was used to remove mobile compounds by treating the samples with a 150 mL 2:1 solution of toluene and ethyl alcohol for 6 h. The resin-extracted samples were then re-dried in a 105 °C oven and ground using a Wiley mill with a 1 mm mesh screen. Following grinding, the homogenized wood tissue was dried in an oven at 60 °C before preparation for isotope ratio mass spectrometry (IRMS), which involved weighing out a sample (1.5 mg $\pm$ 0.15) and enclosing it in a tin capsule.

Carbon isotope discrimination ($\Delta$) was calculated from carbon isotope values of the ground wood samples as follows:

$$\delta_p\ (\text{‰}) = \left( \frac{R_{sample}}{R_{standard}} - 1 \right) \times 1000 \tag{1}$$

$$\Delta = \frac{\delta_a - \delta_p}{1 + \delta_p} \tag{2}$$

where $R_{sample}$ and $R_{standard}$ represent the $^{13}C$ to $^{12}C$ abundance ratios for the samples and the international Vienna Pee Dee Belemnite (PDB) standard, respectively, $\delta_a$ represents $\delta^{13}C$ for source atmospheric $CO_2$ ($-7.9\text{‰}$) and $\delta_p$ indicates the $\delta^{13}C$ for the plant tissue [10,21]. The units are expressed in units of per mil (‰), parts per thousand. WUE values are negatively correlated with values for $\Delta$, so discrimination values close to zero indicate high WUE. All samples were analyzed using a Carlo Erba Elemental Analyzer (Carlo Erba, Milan, Italy) with a zero-blank autosampler at the Duke Environmental Stable Isotope Laboratory in Durham, North Carolina. Accuracy and precision of $\delta^{13}C$ in wood measurements were tested by making repeated measurements of the U.S. National Bureau of Standards NBS-22 (graphite) with a known $\delta^{13}C$ value of $-29.9\text{‰}$ on the PDB scale [22] and at least 2 internal standards (sucrose, cellulose, acetanilide, cyclohexanone and United States Geological Survey (USGS) 40 L-glutamic acid). Our analysis is based on 320 assessments of $\Delta$ values (about three per tree). Values of $\Delta$ were determined at a precision of $+/- 0.2$ per mil at 1 standard deviation relative to NBS-22.

Statistical analyses were carried out for carbon isotope discrimination values, DBH and total tree height based on the linear mixed model:

$$Y_{ijkl} = \mu + B_i + F_j + A_k + FA_{jk} + \varepsilon_{ijkl} \tag{3}$$

where, $Y_{ijkl}$—represents the $l^{th}$ observation in the $i^{th}$ block for the $j^{th}$ family and the $k^{th}$ age/period; $\mu$—represents the grand mean; $B_i$—represents the random effect for blocks, i = 1, 2, 3, 4 with expectations ~ N ($0, \sigma^2_B$); $F_j$—represents the fixed effect for families, j = 1, 2, ... , 9; $A_k$—represents the fixed effect for ages, k = 1, 2, 3, 4 or periods, k = 1, 2, 3; $FA_{jk}$—represents the interaction effect between families and ages/periods and $\varepsilon_{ijkl}$—represents the error term with expectations N ($0, \sigma^2_\varepsilon$).

Analyses, based on the above model were conducted for DBH and height as well as their calculated growth rates per year and $\Delta$. Because repeated $\Delta$ measurements (within-subject) were taken from each tree, a first-order autoregressive covariance structure for correlated errors was included in the model. Type III sums of squares were computed and hypotheses were tested using F tests. Growth rates during ages 7–17, 18–30 and 31–40 for DBH and 7–17, 18–40 and 7–40 for height were determined by subtraction to obtain total growth and then the resulting values were divided by the appropriate number of years. Assumptions of normality were verified visually using histograms and Q-Q plots of residuals as well as comparing means and medians. One tree was excluded from analysis after it was determined to be an outlier for DBH and height, leaving data for 106 trees in our data set. One $\Delta$ sample from the intermediate stage for family 26 $\times$ 32 was found to be contaminated, leaving a total of $n = 317$ values available for analysis. Model terms with significant $p$-values were further tested using the Tukey-HSD multiple comparison test to control the experiment-wise error rate ($\alpha = 0.05$). Pearson

product-moment correlations between pairs of each trait studied were estimated based on individual tree values. All analyses were carried out using SAS 9.4, 2014 (SAS Institute Inc., Cary, NC, USA).

## 3. Results

### 3.1. Height and DBH

Family means for DBH and tree height at ages 7, 17, 30, 40 years showed nonlinear growth trends with less growth occurring at the older ages (Figures 1 and 2, respectively). Height and DBH at age 40 exhibited a moderate, positive correlation ($r = 0.44$, $p < 0.001$). There were significant age, family and family $\times$ age effects for DBH (Table 2). Some families, such as 21 $\times$ 29 and 29 $\times$ 30 tended to have smaller mean DBH values at all ages (Figure 3). Mean DBH ($\pm$ standard error) growth rates during age periods 7–17, 18–30 and 31–40 were 1.1 ($\pm$ 0.02), 0.4 ($\pm$ 0.01) and 0.3 ($\pm$ 0.01) cm/year, respectively and were found to be significantly different from each other (F = 1538.64, $p < 0.001$). DBH at age 40 was significantly correlated with DBH at age 7, 17, 30 with correlation coefficients of $r = 0.35$, 0.83 and 0.95, respectively ($p < 0.001$).

A significant age effect was found for tree height but family and family $\times$ age effects were not statistically significant (Table 2). Mean height growth rate during ages 7–17 was 0.9 m/year and was significantly greater than that found for ages 18–40 when mean growth rate was 0.4 m/year (F = 1623.52, $p < 0.001$). Height at age 40 was moderately correlated with height at age 17 ($r = 0.23$, $p = 0.02$) but not with height at age 7.

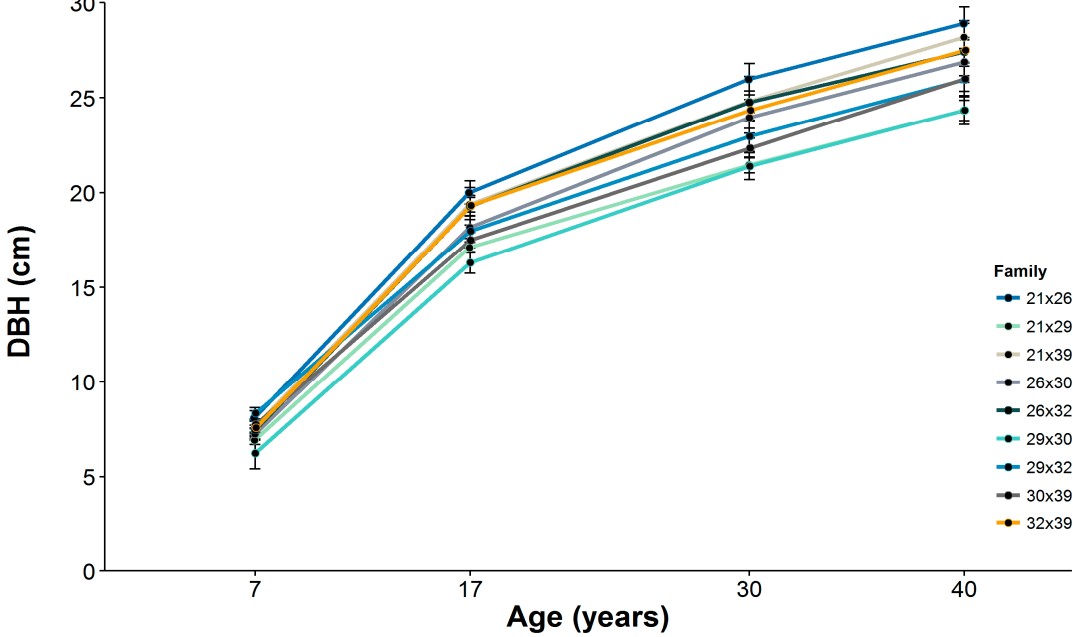

**Figure 1.** Family mean diameter at breast height (DBH) at ages 7, 17, 30 and 40. Vertical lines for each data point represent the standard error of the mean.

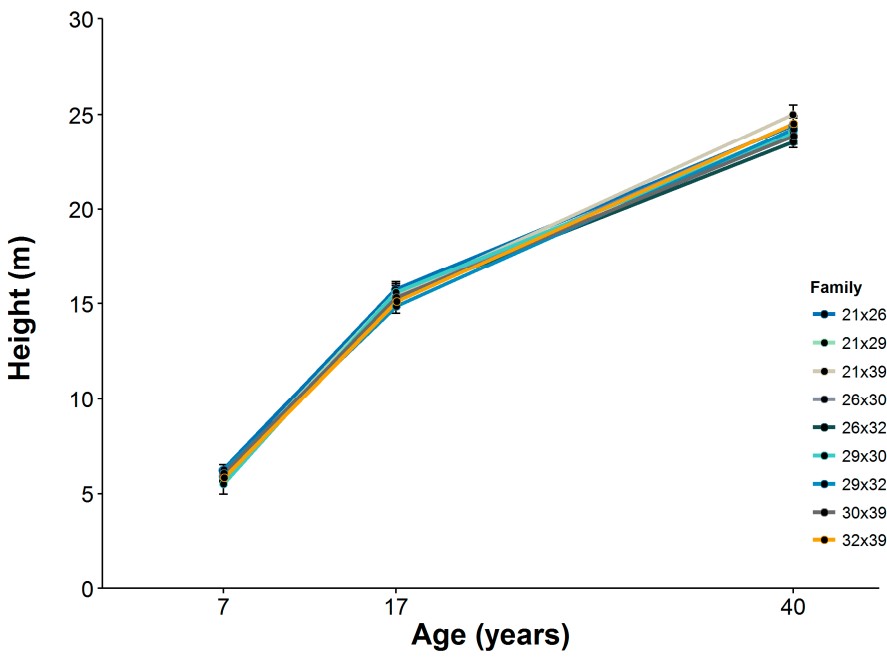

**Figure 2.** Family mean height at ages 7, 17 and 40. Vertical lines for each data point represent the standard error of the mean.

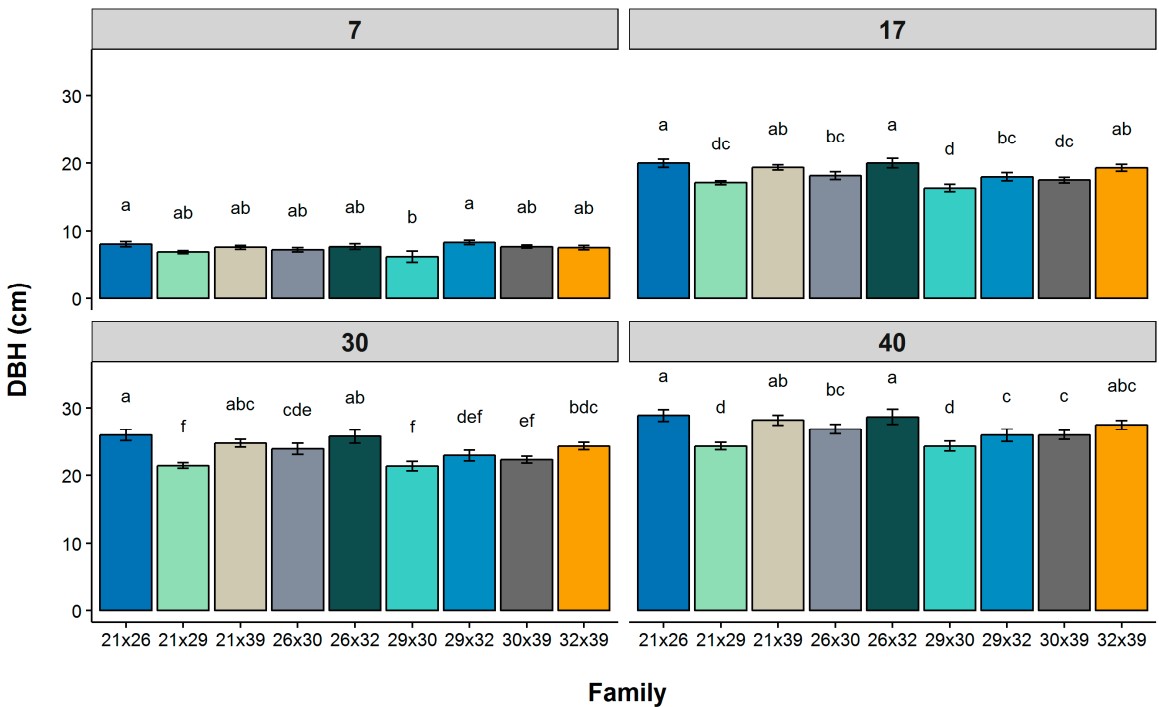

**Figure 3.** Family mean DBH values for all families at ages 7, 17, 30 and 40. Lines on each bar represent the standard error of the mean. Families with the same letter within a time period are not significantly different from each other as indicated by Tukey HSD post-hoc test ($\alpha = 0.05$).

**Table 2.** F test results for DBH, tree height and carbon isotope discrimination (Δ). Values shown are for age, family and family × age (F × A) interaction effects.

| Variable | Numerator DF | Denominator DF | F | Pr |
|---|---|---|---|---|
| **DBH** | | | | |
| Age | 3 | 385 | 4426.51 | <0.001 |
| Family | 8 | 385 | 8.15 | <0.001 |
| F × A | 24 | 385 | 2.98 | <0.001 |
| **Height** | | | | |
| Age | 2 | 288 | 8324.38 | <0.001 |
| Family | 8 | 288 | 0.82 | 0.582 |
| F × A | 16 | 288 | 1.1 | 0.3563 |
| **Δ** | | | | |
| Period | 2 | 287 | 116.41 | <0.001 |
| Family | 8 | 287 | 8.55 | <0.001 |
| F × A | 16 | 287 | 1.73 | 0.0403 |

DF denotes degrees of freedom, Pr denotes probability of the F statistic due to random chance.

### 3.2. Carbon Isotope Discrimination

Observed Δ values for all trees and time periods ranged from 17.2‰ to 20.2‰ and age, family and family × age effects were significant for this trait (Table 2). Mean Δ values (± standard errors (SE)) for age periods 7–17, 18–30 and 31–40 were 18.7‰ (± 0.04), 18.9‰ (± 0.04) and 19.3‰ (± 0.05), respectively. Families 21 × 39, 29 × 32 and 32 × 39 consistently had low mean Δ values for all age periods (Figure 4). Mean Δ generally increased from early to late growth stages, although differences were not statistically significant within all families (Figure 5). Δ values for periods 7–17 and 18–30 were significantly correlated to those observed for ages 31–40 ($r = 0.62$ and 0.79 respectively, $p < 0.001$). Also, Δ values were not significantly correlated with DBH (ages 7–17 $r = 0.03$, $p = 0.76$; ages 18–30 $r = 0.06$, $p = 0.54$; ages 31–40 $r = 0.02$, $p = 0.81$) or height (ages 7–17 $r = 0.18$, $p = 0.06$; ages 18–40 $r = 0.06$, $p = 0.77$) growth rates. Mean Δ values computed over for all time periods were not found to be significantly correlated with either tree DBH values ($r = 0.1$, $p = 0.31$) or tree height values ($r = 0.03$, $p = 0.8$) observed at age 40.

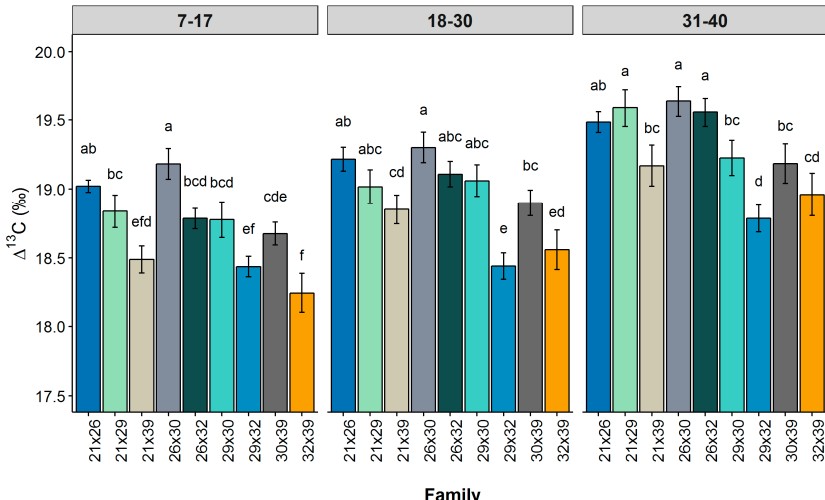

**Figure 4.** Mean (± standard error (SE)) $\Delta^{13}$C (‰) varied significantly among families for early, intermediate and late growth ages. Means with the same letter within each time period are not significantly different from each other as indicated by Tukey HSD post-hoc test ($\alpha = 0.05$).

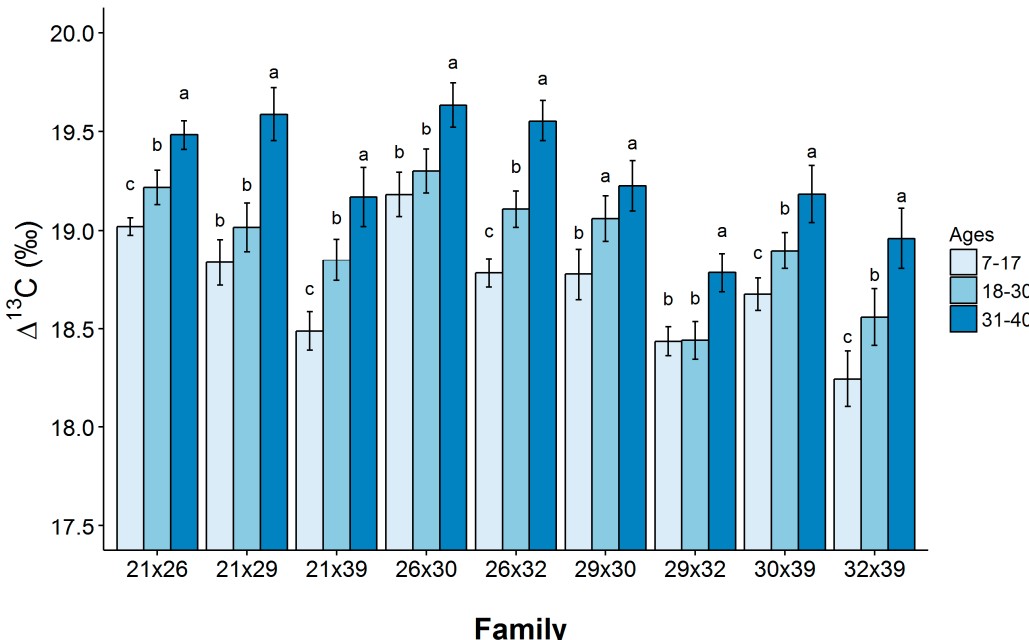

**Figure 5.** Mean $\Delta^{13}C$ (‰) values tend to increase from early to late growth stages within families. Means with the same letter within each family are not significantly different from each other as indicated by Tukey HSD post-hoc test ($\alpha$ = 0.05).

## 4. Discussion

Genetic variation in WUE, as measured by carbon isotope discrimination and growth traits was shown to exist within the subset of longleaf pine full-sib families studied. Our results are consistent with patterns of family variation in growth traits previously reported for longleaf pine [4,23]. Because our sample of families was limited and, because we treated the family effects as fixed in our analysis, we do not make broad conclusions about the overall level of genetic variation in the species [23]. However, the observed genetic variation for growth traits in this analysis agrees with previous evidence, reinforcing the idea that longleaf pine is a species that is suitable for improvement through breeding efforts [24].

Although DBH at age 7 was found to be a fair predictor of DBH at maturity for longleaf pine ($r$ = 0.35), the correlation between DBH at age 17 and 40 increased by more than two-fold to $r$ = 0.83 in 10 years. Thus, more genetic gain in DBH may be achieved by using measurements from age 17 or older ages as a basis for early selection. Such a practice will need to be balanced against the additional time required to complete a breeding cycle. In an analysis of all the families in this test, a strong correlation was found between volume (computed from height and DBH) at ages 17 and 40 but volume at age 7 was not analyzed [23].

Carbon isotope discrimination ($\Delta$) varied significantly among the longleaf pine families studied. Although absolute differences in family means were not large, very small standard errors were associated with $\Delta$ measurements. Minor differences in carbon isotope discrimination can be highly heritable and strongly related to growth [15]. $\Delta$ values for the early and intermediate growth periods were found to be significantly correlated to values obtained for the mature growth stage, indicating it may be advantageous to select trees with higher WUE based on $\Delta$ values observed as early as age 7. Because our results indicate that a significant portion of the phenotypic variation observed in $\Delta$ stems from genetic effects, this suggests there is potential for this trait to be utilized in future longleaf pine tree breeding efforts. Baltunis et al. [12] observed that narrow-sense heritability values for $\Delta$ evaluated in a population of loblolly pine varied across sites, with values of $h^2$ = 0.14, 0.20 and 0.09 in Florida, Georgia and across sites, respectively. In addition, narrow-sense heritability for values of $\Delta$ observed in a population of maritime pine was estimated as $h^2$ = 0.29 [6]. This estimate was obtained from data

collected on trees growing on sites that varied in water availability. Based on these results, it is likely that heritability of Δ in populations of longleaf pine falls within these ranges and there is sufficient additive genetic variance in this trait so that water use efficiency can be increased in populations of this species through breeding.

An age-related decline in WUE is evident from comparison of values for early, intermediate and mature growth stages of longleaf pine. Δ was lowest during the early growth period, which is also the stage in which growth rates were greatest. Nevertheless, we did not find Δ to be correlated with DBH or height growth rates at the family level. An increase ΔWUE was observed across all families and, most likely, was caused by increased competition for light, resulting from canopy closure, and/or decreased availability of water, because of increased root competition.

The observed genetic variation in water-use efficiency in longleaf pine is a possible indication of the ability of some genotypes in this species to be adapted to future environments. Trees in the genus *Pinus* have been found to exercise a drought avoidance strategy, in which water use is decreased quickly under water-limiting conditions and maximized in adequate water environments [25]. Although a comparison of species in the genus *Pinaceae* did not find that longleaf pine has a more conservative water use strategy than other pine species [26], the fact that we detected significant genetic variation for WUE suggests that longleaf pine may be able to adapt to increased, periodic, droughty conditions brought on by climate change, or at least possesses substantial pre-adaptive genetic variation. Similarly, previous research involving maritime pine seedlings found variation in the response of Δ to droughty conditions, indicating genetic differences in phenotypic plasticity as a response to water availability [27].

Although a relationship between growth and water-use efficiency has been observed in other conifer species [6,12,14], such a relationship was not observed in the population of longleaf pine families included in our investigation. Our results suggest it may be possible to select for higher WUE (lower Δ) without affecting growth in longleaf pine. Higher WUE, however, may not necessarily provide trees of this species with a significant growing advantage. Previous research involving longleaf pines found a significant relationship between Δ and tree height at age 5 but only for low productivity sites in Virginia that had not been irrigated [17]. The results we observed were obtained from a test site located in the Gulf Coast region of the U.S. which receives the highest precipitation in the native range of longleaf pine, with up to 1750 mm of rainfall per year [28]. It is possible that a relationship between WUE and tree growth is only found on drier sites where trees are more likely to be water limited. An increase in resolution in future studies could be achieved by identifying drought and/or water-limited years and the corresponding tree rings associated with those years to measure Δ during water limited periods. Such studies would likely provide insight into how WUE responds to water stress and its relationship to growth and/or mortality in trees of longleaf pine.

Harris et al. [29] stressed the importance of considering future climate scenarios in restoration practices and proposes a focus on the balance between restoring historical ecosystems and attempting to build resilient populations for the future. Given the great interest and investment in longleaf pine restoration, it is important to plant appropriate genetic material to ensure the success of restoration efforts. Longleaf pine tree breeding efforts are currently limited as compared to breeding activity in other commercially important southern pine species. In addition to estimating genetic variability for growth traits, it is also important to determine levels of genetic variability in physiological adaptive traits, including water-use efficiency. It is highly important that land managers have access to improved longleaf pine planting stock including that which is best suited for enduring climate-related stressors such as drought.

## 5. Conclusions

This study found significant variation among the families studied for growth in tree diameter as well as in water-use efficiency, as estimated by Δ in mature longleaf pine. Carbon isotope discrimination and tree growth, however, were not found to be correlated. We did observe a pattern of declining

WUE in all families as trees aged. This study is the first to assess Δ in mature longleaf pine and our results provide support for research efforts focused on further characterizing patterns of genetic variation existing for carbon isotope discrimination. In addition to increasing fundamental knowledge of the physiology of this species, our findings provide some insights pertaining to growth trends in longleaf pine that may be useful in tree breeding programs. Moreover, our analysis suggests that, because family differences in diameter growth were detected, it may be possible for tree breeders to independently select for rapid diameter growth and high WUE in longleaf pine.

Identifying patterns of genetic variation occurring in longleaf pine is important because of the considerable current emphasis on planting this species and restoring the longleaf pine ecosystem in the southeastern US. Our study contributes to the research effort that is beginning to address some of the physiological processes under genetic control that may influence longleaf pine's adaptive capacity. Understanding the physiological basis of ecological adaptation in current and future climates is essential for efforts to maintain the vitality of the longleaf pine ecosystem in the future.

**Author Contributions:** A.C.C. designed the experimental protocol, performed the laboratory work, analyzed the results and wrote the manuscript. B.G. and K.H.J. conceived the experiment and advised on the experimental protocol, analysis and writing of the manuscript. C.D.N. and J.H.R. maintained the genetic field trial and supervised logistical and technical aspects of sample collection. All contributing authors participated in the preparation and review of this manuscript.

**Funding:** This work was supported by the United States Department of Agriculture Forest Service [14-CA-11330101-098].

**Acknowledgments:** This research was completed in part thanks to the valuable input during the formulation and analysis of this study by Drs. Fikret Isik, Stacy Nelson and Stephen Kelley. Joshua McRae, Ben Bartlett, Steven Flurry, Chance Parker, Felicia Ogunjube, Karen Sarsony and Kevin Wise assisted with field collection and lab processing.

**Conflicts of Interest:** The authors declare no conflict of interest.

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
