# Peer review of "Genetic Variation in Water-Use Efficiency (WUE) and Growth in Mature Longleaf Pine"

_forests, doi:10.3390/f9110727_

Round 1
Reviewer 1 Report
I enjoyed reviewing this manuscript about genetic variation in water use efficiency in longleaf pine. The long period of height and diameter measurements, and the use of multiple years of wood production for the isotope measurements, are notable strengths of the work. The finding that WUE is not correlated with tree growth is interesting and valuable. My comments and questions are below.
L15: Rather than "seeds" here, I recommend "seedlings" or "trees" because you did not study seeds directly
L18: Resilience refers to capacity to remove from a perturbation. The term "resistance" seems more appropriate here.
L26: Delete "found to be" for brevity
L28: Delete "detected" for brevity
L29: Delete "practice"
L106: More information about the original geographic provenance of the families is needed.
L121: I do not understand "extracted west to east from tree boles." Do you mean that you extracted one core from the east side of the tree and one core from the west side?
L127: Delete comma after "mobile"
Figure 1 and 2 captions: Instert "Vertical" Before "Lines" so that it is clearer that you mean the vertical lines.
L203: Delete "determined to be" for brevity
L208-10: Please list the r and p values for these non-significant correlations so the reader can judge the evidence for the stated lack of association.
L222: Provide a little more information about how the results are consistent with previous reports.
L235-250: Can you calculate heritability for your results and compare to previous reports?
L256: I am surprised that you speculate that decreased water would increase delta C13. I think the response to drier soil would be a decrease in delta C13 (less discrimination, increase WUE) because stomatal conductance usually declines during drought. Please reconsider this issue.
L290: Delete comma after "that"
Author Response
Responses to Reviewer 1—Manuscript Forests 388655
The authors wish to thank Reviewer#1 for the careful, comprehensive and constructive review. After each reviewer comment is our response, marked in italics and red font color.
Reviewer comments and responses:
I enjoyed reviewing this manuscript about genetic variation in water use efficiency in longleaf pine. The long period of height and diameter measurements, and the use of multiple years of wood production for the isotope measurements, are notable strengths of the work. The finding that WUE is not correlated with tree growth is interesting and valuable. My comments and questions are below.
L15: Rather than "seeds" here, I recommend "seedlings" or "trees" because you did not study seeds directly-----Yes
L18: Resilience refers to capacity to remove from a perturbation. The term "resistance" seems more appropriate here.—Chose to replace resilience with survival and growth in a changing climate.
L26: Delete "found to be" for brevity----Yes
L28: Delete "detected" for brevity---Yes
L29: Delete "practice" ---Yes
L106: More information about the original geographic provenance of the families is needed.---Yes, this has been clarified to state that the parents were from the same locale of the test.
L121: I do not understand "extracted west to east from tree boles." Do you mean that you extracted one core from the east side of the tree and one core from the west side?---The methods have been clarified.
L127: Delete comma after "mobile"----We disagree with removing the comma, as “more mobile” is a phrase that can be removed from the sentence without changing the meaning entirely.
Figure 1 and 2 captions: Instert "Vertical" Before "Lines" so that it is clearer that you mean the vertical lines.--Yes
L203: Delete "determined to be" for brevity----Yes
L208-10: Please list the r and p values for these non-significant correlations so the reader can judge the evidence for the stated lack of association.—Yes, r and P values have been added to the text.
L222: Provide a little more information about how the results are consistent with previous reports.---Yes, this statement has been clarified.
L235-250: Can you calculate heritability for your results and compare to previous reports?---No, the authors thoroughly discussed if it was appropriate to calculate heritability for these traits, but decided that such a calculation on so small a genetic sample would be potentially misleading.
L256: I am surprised that you speculate that decreased water would increase delta C13. I think the response to drier soil would be a decrease in delta C13 (less discrimination, increase WUE) because stomatal conductance usually declines during drought. Please reconsider this issue.---Yes, changed to “increase in WUE.”
L290: Delete comma after "that"----Yes

Reviewer 2 Report
This paper compares growth, and carbon isotope discrimination across longleaf pine genetic families from tree cores taken from 40 year old trees. While the data presented is fairly standard, it is useful in continuing a breeding program for longleaf pine, a species that is the focus of restoration efforts in the Southeastern US. Some more indepth analysis could be done if monthly or yearly precipitation data were available to compare with growth ring widths and carbon isotope data. Also, the acronyms used for carbon isotope discrimination are confusing. The authors use Δ13C throughout the text to mean carbon isotopic "discrimination". Other literature (ie Farquhar et al, 1989 Carbon isotope discrimination and photosynthesis) use δ13C for the ratio of carbon isotopes in plant material vs. the standard (-1) and capital delta, Δ, as a calculated parameter called "discrimination" And this is how it is presented in the equations in the methods of this manuscript. Therefore, it might be clearer to not use the delta's and just call it carbon isotope discrimination. More specific comments as follows:
Minor note but some of the "keywords" are also found in the title, but to use words or phrases that are not in the title.
Introduction:
Line 41: perhaps specify you are referring to southern pines here
Line 55: Intrinsic water use efficiency is calculated as photosynthesis/stomatal conductance
Line 56: again be specific about capital delta here, it has a specific formula and is different from lower case delta 13C
Line 57: a more important point here is that during low stomatal conductance conditions (low internal CO2) Rubisco is forced to use 13C for photosynthesis and discrimination is lower
Line 64-65: unless stored carbon is being use for wood production, then there can be lags in the signal
Materials and Methods
Line 98: what spacing were they planted at, provide general info about how the sites were managed for competition etc.
Line 138: these equations provide the proper use of lower case vs. capital delta
Line 140: how did you measure the atmospheric CO2 δ13C or is this estimate pulled from the literature? This is important because the atmospheric signal can change (and likely has changed) over your study period which will affect these results.
Line 141: high discrimination means plenty of CO2 in the leaf from high stomatal conductance, high stomatal conductance = low water use efficiency (at the same photosynthetic rate) therefore discrimination is inversely correlated with WUE
Results
Line 201 and beyond: this might read better if you just referred to Δ13C as "discrimination"
Discussion:
Line 256: If they are less water use efficient when they are older, they likely have more access to water, through developing a deeper taproot, canopy closure decreasing vapor pressure deficit at the leaf level etc
Line 271: Period instead of the first comma
Author Response
The authors wish to thank Reviewer#2 for the careful, comprehensive and constructive review. After each reviewer comment is our response, marked in italics and red font color.
Reviewer#2 comments:
This paper compares growth, and carbon isotope discrimination across longleaf pine genetic families from tree cores taken from 40 year old trees. While the data presented is fairly standard, it is useful in continuing a breeding program for longleaf pine, a species that is the focus of restoration efforts in the Southeastern US. Some more indepth analysis could be done if monthly or yearly precipitation data were available to compare with growth ring widths and carbon isotope data. Also, the acronyms used for carbon isotope discrimination are confusing. The authors use Δ13C throughout the text to mean carbon isotopic "discrimination". Other literature (ie Farquhar et al, 1989 Carbon isotope discrimination and photosynthesis) use δ13C for the ratio of carbon isotopes in plant material vs. the standard (-1) and capital delta, Δ, as a calculated parameter called "discrimination" And this is how it is presented in the equations in the methods of this manuscript. Therefore, it might be clearer to not use the delta's and just call it carbon isotope discrimination. The acronym issue raised by the reviewer has been addressed. In the abstract and introduction we define the acronym for “carbon isotope discrimination” as “Δ” and have substituted this term for the Δ13C, except for the equations in the methods and in two of the figures where the precise term was needed for the axis of the graphs.
More specific comments as follows:
Minor note but some of the "keywords" are also found in the title, but to use words or phrases that are not in the title.----Yes, corrected
Introduction:
Line 41: perhaps specify you are referring to southern pines here—Clarified to make it clear we are referring specifically to longleaf pine here, as there is quite a bit of research on this topic for other southern pines.
Line 55: Intrinsic water use efficiency is calculated as photosynthesis/stomatal conductance---Yes, this language has been corrected.
Line 56: again be specific about capital delta here, it has a specific formula and is different from lower case delta 13C
Line 57: a more important point here is that during low stomatal conductance conditions (low internal CO2) Rubisco is forced to use 13C for photosynthesis and discrimination is lower---Yes, this language has been clarified.
Line 64-65: unless stored carbon is being use for wood production, then there can be lags in the signal—Yes, this possibility has been acknowledged in the text.
Materials and Methods
Line 98: what spacing were they planted at, provide general info about how the sites were managed for competition etc.---Yes, this information has been added.
Line 138: these equations provide the proper use of lower case vs. capital delta
Line 140: how did you measure the atmospheric CO2 δ13C or is this estimate pulled from the literature? This is important because the atmospheric signal can change (and likely has changed) over your study period which will affect these results.---As stated in the text, we used a literature value of -7.9 ‰, as stated in the references cited.
Line 141: high discrimination means plenty of CO2 in the leaf from high stomatal conductance, high stomatal conductance = low water use efficiency (at the same photosynthetic rate) therefore discrimination is inversely correlated with WUE—This language has been clarified, by substituting the term “discrimination” for the term “results.”
Results
Line 201 and beyond: this might read better if you just referred to Δ13C as "discrimination"---Yes, please see the note above about the use of the Δ to represent the term carbon isotope discrimination.
Discussion:
Line 256: If they are less water use efficient when they are older, they likely have more access to water, through developing a deeper taproot, canopy closure decreasing vapor pressure deficit at the leaf level etc---Yes, changed to “increase in WUE.”
Line 271: Period instead of the first comma---Yes